# Molecular Inverse Comorbidity between Alzheimer’s Disease and Lung Cancer: New Insights from Matrix Factorization

**DOI:** 10.3390/ijms20133114

**Published:** 2019-06-26

**Authors:** Alessandro Greco, Jon Sanchez Valle, Vera Pancaldi, Anaïs Baudot, Emmanuel Barillot, Michele Caselle, Alfonso Valencia, Andrei Zinovyev, Laura Cantini

**Affiliations:** 1Department of Physics and INFN, Universitá degli Studi di Torino, via P. Giuria 1, 10125 Turin, Italy; a.greco@dkfz-heidelberg.de (A.G.); caselle@to.infn.it (M.C.); 2Institut Curie, PSL Research University, 75005 Paris, France; emmanuel.barillot@curie.fr (E.B.); andrei.zinovyev@curie.fr (A.Z.); 3INSERM U900, 75248 Paris, France; 4CBIO-Centre for Computational Biology, Mines ParisTech, PSL Research University, 75006 Paris, France; 5Aix Marseille Université, INSERM, MMG, CNRS, 13005 Marseille, France; 6Centre de Recherches en Cancérologie de Toulouse (CRCT), UMR1037 Inserm, ERL5294 CNRS, 2 Avenue Hubert Curien, 31037 Toulouse, France; 7Université de Toulouse, Université Toulouse III Paul Sabatier, 31062 Toulouse, France; 8Barcelona Supercomputing Center (BSC), 08034 Barcelona, Spain; jon.sanchez@bsc.es (J.S.V.); vera.pancaldi@inserm.fr (V.P.); anais.baudot@univ-amu.fr (A.B.); alfonso.valencia@bsc.es (A.V.); 9ICREA, 08010 Barcelona, Spain; 10Computational Systems Biology Team, Institut de Biologie de l’Ecole Normale Supérieure, CNRS UMR8197, INSERM U1024, Ecole Normale Supérieure, Paris Sciences et Lettres Research University, 75005 Paris, France

**Keywords:** networks, Alzheimer’s disease, lung cancer, inverse comorbidity, transcriptome, matrix factorization

## Abstract

Matrix factorization (MF) is an established paradigm for large-scale biological data analysis with tremendous potential in computational biology. Here, we challenge MF in depicting the molecular bases of epidemiologically described disease–disease (DD) relationships. As a use case, we focus on the inverse comorbidity association between Alzheimer’s disease (AD) and lung cancer (LC), described as a lower than expected probability of developing LC in AD patients. To this day, the molecular mechanisms underlying DD relationships remain poorly explained and their better characterization might offer unprecedented clinical opportunities. To this goal, we extend our previously designed MF-based framework for the molecular characterization of DD relationships. Considering AD–LC inverse comorbidity as a case study, we highlight multiple molecular mechanisms, among which we confirm the involvement of processes related to the immune system and mitochondrial metabolism. We then distinguish mechanisms specific to LC from those shared with other cancers through a pan-cancer analysis. Additionally, new candidate molecular players, such as estrogen receptor (ER), cadherin 1 (CDH1) and histone deacetylase (HDAC), are pinpointed as factors that might underlie the inverse relationship, opening the way to new investigations. Finally, some lung cancer subtype-specific factors are also detected, also suggesting the existence of heterogeneity across patients in the context of inverse comorbidity.

## 1. Introduction

Large-scale genomics projects, including The Cancer Genome Atlas (TCGA, https://www.cancer.gov/tcga), are currently providing an overwhelming amount of omics data. The available data offer the opportunity to better understand biological systems and cancer in particular, but their high dimensionality poses considerable challenges typical of “Big Data” [1].

A powerful approach to analyze and interpret large datasets is represented by matrix factorization (MF), a class of unsupervised methods that reduces high-dimensional data into low dimensional subspaces, while preserving as much information as possible [2,3,4]. Given a data matrix X, representing measures of the expression of genes across a set of samples, MF learns two sets of low-dimensional representations: “metagenes”, encoding molecular relationships, and “metasamples”, encoding sample-level relationships. Up until now, MF has been successfully used in a broad spectrum of applications: unsupervised clustering, especially in the context of cancer subtyping [5,6], molecular pattern discovery [7,8], mutational signatures definition [9,10] and tumor sample immune infiltration quantification [11]. Such results have been obtained by mining single large-scale datasets with MF, such as the transcriptome or methylome. Recently, we designed a metric to infer univocal correspondences between the metagenes obtained by an MF algorithm on multiple independent datasets profiled from the same biological condition (e.g., the same cancer tissue), and used this metric to design a methodological framework that revealed relevant pathways characteristic of colorectal cancer [12].

Here, we are interested in investigating the molecular bases of previously documented disease–disease (DD) relationships. Indeed, several computational studies have inferred DD relationships, starting from the “Human Disease Network” where diseases were connected when sharing disease genes [13], and more recently with the “multiplex network of human diseases” in which genotype- and phenotype-based network layers were used to propose new disease-associations [14]. More importantly, DD relationships have also been systematically identified by epidemiological studies, working at the level of populations and looking for the co-occurrence of different diseases in the same patients using medical claims [15], medical records [16] and insurance claims [17]. The higher than expected risk of developing pancreatic cancer in patients suffering from type II diabetes [18] and the higher susceptibility to lung cancer in asthma patients [19] are among the most renowned examples of cancer-related comorbidities. Interestingly, it has also been described that patients suffering from certain diseases have a lower than expected risk of developing specific cancers, a phenomenon known as inverse comorbidity [20,21,22]. An example of these protective effects of one disease against another is represented by the documented inverse comorbidity between Alzheimer’s disease (AD) and lung cancer (LC) [22,23,24]. Molecular and non-molecular factors (e.g., the environment, lifestyle or drug treatments) can be responsible for such DD relationships. The molecular mechanisms underlying these DD relationships are poorly understood and investigating them offers unprecedented opportunities to better understand the etiology and pathogenesis of diseases, with the hope of identifying opportunities for repositioning pre-existing treatments.

Recently, transcriptomic meta-analyses revealed sets of significantly up- and down-regulated genes that are shared across diseases displaying different patterns of direct and inverse comorbidities [25,26]. We hereby propose to use an MF approach to study the molecular bases of DD relationships. In fact, previous methods based on differential expression analysis only focus on the predominant signals present in the data, failing to capture alternative signals and local behaviors [3]. These limitations are overcome by MF, which allows us to learn metagenes, i.e., rankings of genes, without focusing on single sets of predominant genes. Moreover, contrary to differential expression analysis, MF jointly provides metagenes and metasamples, as it also groups samples together in a way that can be related to their biological characterization. Despite the great potential of MF in this context, applying this methodology to the study of the molecular bases of DD relationships requires some specific modifications. We thus extend our previously defined MF framework for the particular study of DD relationships [12]. Moreover, given the existence of positive and negative DD connections, we also adapt the framework to distinguish molecular relationships concordantly and discordantly altered in datasets coming from different diseases.

Considering the inverse comorbidity between Alzheimer’s disease (AD) and lung cancer (LC) as a case study [22,23,24], we applied our MF framework to 17 transcriptomic datasets, including both LC and AD samples (a total of 1367 samples), and we highlighted multiple molecular mechanisms possibly underlying the inverse comorbidity pattern. Through a pan-cancer analysis, we categorized the processes that we found to be involved in AD–LC inverse comorbidity according to their presence in other cancers. The previously identified role of the immune system and mitochondrial metabolism in AD–LC inverse comorbidity is confirmed by our analysis. Additionally, new candidate molecular players, such as estrogen receptor (ER), CDH1 and histone deacetylase (HDAC), are identified as potentially involved in the inverse comorbidity considered. Finally, some lung cancer subtype-specific alterations are also detected, suggesting the existence of heterogeneity across patients in the context of inverse comorbidity.

## 2. Results

### 2.1. A New MF Framework to Study Disease–Disease Relationships

We previously defined the reciprocal best hit (RBH), a metric to infer univocal correspondences between the MF metagenes obtained on independent transcriptomic datasets measured from the same biological condition (e.g., same cancer tissue) [12]. Based on this metric we designed an RBH-based framework, which we have separately applied to three large collections of transcriptomic datasets covering a total of 10,490 patients. The method involves three sequential steps: (1) each transcriptomic dataset is independently decomposed into metagenes and metasamples with MF; (2) using the RBH metric, links between metagenes are inferred and a network is constructed; (3) communities of metagenes are detected in the RBH network. These communities of metagenes are then analyzed for functional relatedness and provide a biological interpretation of the principal factors that shape the transcriptomes. Here, we adapted the RBH-based framework to the study of the molecular mechanisms underlying DD relationships in order to infer univocal positive/negative correspondences between MF metagenes independently obtained on datasets measured from different diseases. The improved RBH-based framework proposed here maintains the three sequential steps described above. At the same time, the following methodological novelties are introduced in each of the steps defined above: (i) we propose two methodologies for the orientation of the metagenes (i.e., we assign a sign to the metagenes in order to express either direct or inverse similarity between them); (ii) we derive a novel definition of the reciprocal best hit (RBH) network taking into account the orientation of the metagenes; and (iii) we restrict the community detection phase to the subnetwork of interest (e.g., subnetwork of negative links connecting metagenes of LC and AD in our case). The structure of the framework, together with the changes here introduced are summarized in Figure 1. The code to run the framework is available at https://github.com/agreco92/inverse_comorbidity_MF.

#### 2.1.1. Step 1: Data Decomposition and Orientation of the Components

Each transcriptomic dataset is separately decomposed using MF. The framework proposed here can be combined with any MF algorithm of interest. In this work, we chose stabilized independent component analysis (sICA) [12,27,28], a stabilized version of ICA [28,29,30]. sICA was indeed previously shown to outperform alternative MFs in the extraction of relevant biological knowledge from collections of transcriptomic datasets derived from the same biological condition (e.g., the same cancer type) [12]. Moreover, the ability of sICA to separate the various overlapping biological factors present in transcriptomic data, such as those linked to tumor cells, the tumor microenvironment, non-biological factors, sample processing or data generation, makes this approach particularly promising for extracting relevant molecular factors from the numerous confounding factors involved in DD relationships.

By applying sICA to a transcriptomic matrix *X (n x m)*, with *n* genes in the rows and *m* samples in the columns, we reduce it to the product of an unknown mixing matrix *A (n x k)*, whose columns are here denoted as “metagenes” and an unknown matrix of source signals *S (k x m)*, whose rows are here denoted as “metasamples”. The metagene/metasample associated to component *i* will thus provide the contribution of each gene/sample present in matrix *X* to component *i*. Metasamples and metagenes are learned based upon the assumption that the number *k* of components occurring in the input matrix *X* is smaller than either its rows or columns. Here, we selected the number *k* of components to be 100 for those datasets with more than 100 samples and equal to half of the samples for smaller datasets. These chosen values of *k* are higher than the estimation of the optimal transcriptomic dimension (optimal *k* value defined in [28]), due to the fact that overdecomposition in sICA was proven not to be detrimental for the interpretability of the resulting components [28].

To determine the orientation of the sICA metagenes, two alternative approaches are considered: “long tail-pointing” and “disease-pointing”. As shown in Figure 2A, for each metagene, the “long tail-pointing” approach checks the distribution of its weights; if the long tail of such distribution corresponds to the positive side of the metagene, no further operation is performed. If instead the long tail of the metagene distribution corresponds to negative weights, then we flip the metagene and its associated metasample. The long tail-pointing approach has been previously used for other sICA applications [28]. The reason for choosing such an approach is that sICA metagenes are identified by maximizing non-Gaussianity of the data projections. As a consequence, the longest-tails of such distributions are those containing most of the biological information.

As an additional approach for the orientation of the sICA metagenes, we introduce the “disease-pointing” method (Figure 2B). As shown in Figure 2B, the “disease-pointing” approach evaluates the orientation of a metagene based on the distribution of the weights of its associated metasample. A Wilcoxon test is applied to the weights of the metasample to test its differential association to case vs. control. If the test statistic obtained from the Wilcoxon test is positive, no further operation is performed. If, instead, the Wilcoxon test statistic is negative, we flip the metasample and its associated metagene.

#### 2.1.2. Step 2: Construction of the Reciprocal Best Hits (RBHs)

At Step 2, the reciprocal best hit (RBH) network is constructed. A positive/negative RBH is defined as follows: given two sets of metagenes $\left\{M_{1}\ldots .M_{k}\right\}$ and $\left\{N_{1}\ldots .N_{k}\right\}$ obtained from the transcriptomic datasets $T^{M}$ and $T^{N}$, respectively, we define $M_{i}$ and $N_{j}$ as a positive reciprocal best hit (+RBH) if:
(1)$$max\left(cor\left(M_{i},{\left\{N_{t}\right\}}_{t=1}^{k}\right)\right)\;=max\left(cor\left({\left\{M_{t}\right\}}_{t=1}^{k},N_{j}\right)\right)>0$$

and $M_{i}$ and $N_{p}$ as a negative reciprocal best hit (–RBH) if:
(2)$$min\left(cor\left(M_{i},{\left\{N_{t}\right\}}_{t=1}^{k}\right)\right)=min\left(cor\left({\left\{M_{t}\right\}}_{t=1}^{k},N_{p}\right)\right)<0$$where *cor* corresponds to the Spearman correlation.

Briefly, as defined in Equations (1) and (2), the two metagenes $M_{i}$ and $N_{j}$ are linked by a ±RBH if the absolute value of their correlation is maximal with respect to both the correlations of with all other metagenes obtained from dataset $T^{N}$ and the correlations of $N_{j}$ with all other metagenes obtained from dataset $T^{M}$. For defining a +RBH (Equation (1)), positive correlations are considered, while for –RBH (Equation (2)), negative correlations are taken into account. As a result, each metagene will thus find a maximum of two associated metagenes in another independent transcriptomic dataset, corresponding to +RBH Equation (1) and –RBH Equation (2). Repeating the same procedure for the metagenes of all the available transcriptomes, we obtain a weighted network whose nodes are the metagenes computed in all the transcriptomic datasets and whose links correspond to their +RBHs and −RBHs computed as in Equations (1) and (2). These links are weighted based on the correlation computed in Equations (1) and (2). The procedure used to construct the ±RBHs preferentially builds a sparse network with high weights; however, a cutoff threshold can be always introduced on the weights if required.

#### 2.1.3. Step 3: Subnetwork Isolation and Community Detection

In Step 3, given our interest in the processes that are differentially altered between two diseases such as AD and LC, we delineate the relevant subnetwork of RBHs. For example, if we want to study the inverse comorbidity between AD and LC, we restrict the analysis to the negative RBHs (−RBHs) connecting metagenes of AD with metagenes of LC. Once the subnetwork of interest is selected, we detect communities in this subnetwork with the MCL algorithm [31,32]. When running MCL, the weights associated to the links of the RBH network are taken into account for community detection. The obtained communities correspond to biological components involved in the DD relationship and consistently present in multiple independent datasets. Moreover, having previously isolated the subnetwork of interest (such as negative RBHs between AD and LC), we are sure to only identify communities that are altered in the direction consistent with the type of comorbidity under analysis (oppositely regulated in the case of inverse comorbidity and concordantly regulated in the case of direct comorbidities). The obtained communities are then biologically annotated and interpreted as described in the Methods section.

### 2.2. Investigation of the Orientation Methodology for the sICA Components

Among the various modifications made to the framework, of particular importance is the choice of the procedure for the orientation of the metagenes. As described previously, two alternative approaches were considered: “long tail-pointing” and “disease-pointing”. We tested how such a choice impacts the following steps of the framework and, in particular, the structure of the obtained RBH network. To do this, we selected a specific DD relationship, i.e., the inverse comorbidity between Alzheimer’s disease (AD) and lung cancer (LC) as a case study [22,23,24].

We considered 17 transcriptomic datasets, spanning AD and LC patients and containing case and control samples, (see the Methods section for further details). Following our framework (Figure 1), each dataset was decomposed separately through sICA (see Appendix A for the number of components) and the orientation of the components was established both with the long-tail-pointing and the disease-pointing approaches. The resulting metagenes were then compared according to multiple criteria (see Figure 3).

First, the correlation between the obtained metagenes and the case vs. control fold-change of expression was considered to decide between the two orientation methods. Fold change measures expression changes between two states (case vs. control in our case) and for each gene in the log2-transformed expression matrix, it is computed as Avg(Case samples)−Avg(Control samples). In fact, to associate a metagene to a specific biological function or pathway, we need to perform enrichment tests using databases of functional annotations (e.g., Reactome, Gene Ontology). Generally, this interpretation step is just aimed at associating a function to each metagene without considering the sign of activity of the identified pathways/processes. However, when dealing with comorbidities, it is important to not only associate a function to each metagene, but also to infer the sign of activity of such pathways/functions. This task can be easily achieved once the metagenes are positively correlated with the gene fold-change. As shown in Figure 3A, the disease-pointing orientation produces metagenes that are significantly more correlated with the gene fold-change than the long tail-pointing one (significance tested with Wilcoxon test, resulting *p*-values available in Appendix A).

We have then applied Step 2 of the framework and independently constructed an RBH network for “long-tail-pointing” and “disease-pointing” oriented metagenes. In both cases, the nodes of the network correspond to the metagenes independently identified in the 17 datasets (369 total nodes) and their links are +/–RBHs, defined as in Equations (1) and (2). Changes in the orientation of the metagenes alter the sign of the correlations, giving rise to different RBH networks. We have thus compared the “long-tail-pointing” vs. “disease-pointing” RBH networks based on their number of links (Figure 3B). The “disease-pointing” method returns 1616 RBHs vs. the 1574 returned by the “long-tail-pointing” method. Such a result is due to the higher number of –RBHs identified with the “disease-pointing” orientation (802 vs. 705).

In Step 3, we focused on the subnetwork composed of –RBHs and linking AD components with LC ones and vice-versa, which in the following we call “–RBH AD/LC subnetwork”. These are in fact the metagenes and RBHs of interest for the study of the AD–LC inverse comorbidity. We studied the topology of this subnetwork starting from its number of nodes and links (Figure 3C). The –RBH AD/LC subnetwork based on the “disease-pointing” orientation includes a higher number of metagenes (167 vs. 127 of “long-tail-pointing”) and a higher number of links (268 vs. 194 of “long-tail-pointing”). Moreover, among the RBHs present in the subnetwork, those of the “disease-pointing” tend to be more frequently connecting factors that are significantly differential between case and control (112 vs. 70 of “long-tail-pointing”). Communities were then detected in the “long-tail-pointing” and “disease-pointing” –RBH AD/LC subnetworks. As shown in Figure 3D, the “disease-pointing” –RBH AD/LC subnetwork has a higher modularity (0.49 vs. 0.43) and higher clustering coefficient (0.49 vs. 0.39). Moreover, 20 communities of size higher or equal to four are detected in the “disease-pointing” –RBH AD/LC subnetwork vs. the 12 of the alternative approach and the average size of the “disease-pointing” communities is 4.3 vs. the 4.2 of the alternative approach (Figure 3E). An overview of all the communities obtained in the “disease-pointing” –RBH AD/LC subnetwork can be found in Appendix A.

Overall, our analysis indicates that the “disease-pointing” orientation tends to identify a higher number of candidate molecular processes/pathways involved in AD–LC inverse comorbidity. For all these reasons, “disease-pointing” is the orientation approach that we selected for the following analysis.

### 2.3. New Biological Insights on the Inverse Comorbidity between AD and LC

We hypothesize that the communities of the –RBH AD/LC subnetwork obtained with the “disease-pointing” orientation could be related to the AD–LC inverse comorbidity. We thus annotated the communities of the –RBH AD/LC subnetwork by using MsigDB signatures [33], microenvironment cell population (MCP) counter signatures [34], predefined lung cancer subtypes [35] and the metagenes computed in [27], here referred to as CIT, as described in the Methods section. The obtained –RBH AD/LC subnetwork with the biological annotations is illustrated in Figure 4 and Appendix A.

The majority of the communities present in the network are associated to the immune system and mitochondrial functioning, confirming the results of previous transcriptomics meta-analyses on the inverse comorbidity between AD and LC [25,26]. Interestingly, these processes are clearly partitioned into multiple communities, suggesting that we can detect more precisely which biological processes are involved in the inverse comorbidity. Specifically, fibroblasts, neutrophils, monocytes, B cells and T cells are the immune cells showing an inverse activity in LC and AD according to our analysis. Moreover, communities involved in the regulation of two immune-system related drugs (fenretinide and corticosteroids) are identified. Interestingly, corticosteroid consumption is associated with a lower risk of Alzheimer’s neuropathology [36], while their use in LC patients is associated with lower overall survival [37]. At the same time, fenretinide has been shown to inhibit growth in lung cancer cell lines [38] and it has been proposed as a potential adjuvant for late onset AD [39]. Moreover, the fenretinide community is tightly linked with the monocytes one, in agreement with its mechanisms of action involving the regulation of the secretion of pro-inflammatory cytokines in human monocytes [40].

The communities associated to mitochondria span different processes related to their activity: oxidation-reduction processes (communities 16, 23, 40), hypoxia (community 10) and phosphate metabolic processes (community 38). Enrichment in hypoxia could correspond to a confounding factor linked to the state of the profiled tissues (post-mortem for AD and fresh tissue biopsy for LC). However, patients suffering from systemic or prenatal hypoxia have a higher risk of developing Alzheimer’s disease [41,42] and targeting hypoxia seems to improve lung cancer outcome [43], indicating that the hypoxia-related community could also contain medically relevant information.

Additionally to mitochondria and the immune system, community 36 has been associated to gender, in line with the higher risk of females to develop Alzheimer’s disease, in opposition to lung cancer, which is more frequent in men [44,45]. Histone deacetylase (HDAC), associated to community 22, confirms the known involvement of HDAC1 in both cancer and Alzheimer’s disease [46,47]. Community 30 is enriched in focal adhesion. The inhibition of focal adhesion kinase, which is overexpressed in several cancers, decreases cell viability [48], while in the case of Alzheimer’s disease, amyloid-ß induces the inactivation of focal adhesion kinase [49]. Cell cycle and CDH1 targets are associated to community 24. Interestingly, growing evidence suggests that dysregulation of APC/C-CDH1 is involved in neurodegenerative diseases, potentially as a consequence of amyloid-β driven proteasome-dependent degradation of CDH1 [50]. On the other hand, a significantly higher methylation level of CDH1, inducing its inactivation, plays an important role in lung cancer [51]. Community 20 is associated to protein processing and chaperone-mediated protein folding. Protein misfolding is a known marker of AD [52,53]. At the same time, cell division, migration, and invasion rely on microtubules and actin filament components and thus chaperone-mediated protein folding activity is tightly linked to cancer [54]. Similar arguments support the involvement of microtubules (community 6) to AD–LC inverse comorbidity. Moreover, response to estrogen receptor (ER) (“ESR1 targets”) has been found to be enriched in 20 communities, but without a clear association to a specific one. Interestingly, an inverse association has been shown between the use of estrogen and early onset of Alzheimer’s disease, suggesting that it might have protective effects against the disease [55], potentially due to its inhibitory activity on neuroinflammation [56]. On the other hand, the use of hormonal replacement therapy significantly increases LC mortality, supporting an opposite role of estrogen in lung cancer [57]. These inverse effects of regulation of focal adhesion, CDH1 and estrogen receptor in cancer and AD are consistent with a possible association of these pathways to the inverse comorbidity patterns observed.

We then explored if lung cancer subtype-specific molecular mechanisms could also be involved in the AD–LC inverse comorbidity [35]. The main biologically-annotated network communities were found to be generic of LC with no association to a specific subtype. However, three communities in our network (27, 14 and 3) mapped to the three predefined LC subtypes (proximal proliferative, proximal inflammatory and terminal respiratory unit, respectively). Therefore, some lung cancer subtype-specific regulatory programs seem to also be involved, suggesting the existence of across-patient heterogeneity, even if such a phenomenon is not the predominant one.

Finally, AD has been shown to have comorbidity relationships at the epidemiological level, not only with LC, but also with other cancer types, with most of them being inverse comorbidities [13]. We thus tested if some of the candidate biological processes that we have identified to be possibly involved into AD–LC inverse comorbidity could be generalized to the comorbidity relationship between AD and other cancers. With this aim, we considered metagenes previously computed on TCGA transcriptomes for 32 different cancer types (pan-cancer metagenes) [28] and inferred their RBHs with the metagenes of the –RBH AD/LC subnetwork. The presence of pan-cancer metagenes in the communities of the –RBH AD/LC subnetwork was then tested. If a community in the –RBH AD/LC subnetwork is found to be correlated with some pan-cancer metagenes, we can infer that such processes could also have a role in the relationship between AD and other cancers. We thus quantified the number of the connected pan-cancer metagenes for each community of the –RBH AD/LC subnetwork (see Appendix A for results). Of note, the orientation of the TCGA pan-cancer metagenes has not been defined in [28]; we thus cannot infer here if the activity of the pan-cancer metagenes is concordant with that of the LC metagenes or of the AD ones.

As reported in Appendix A, the majority of the identified communities, corresponding to the immune system-related signals, gender, chrX and mitochondrial activity, matches metagenes obtained from other cancers, indicating a possible role of such processes in the co-morbidity of AD with other cancers. On the other hand, five communities (19%), corresponding to LC subtypes, IFN-γ and phosphate metabolism, are found to be specific to AD–LC inverse comorbidity.

## 3. Discussion

Matrix factorization (MF) is a prominent solution for high-dimensional omics data analysis with a vast range of applications in computational biology.

Here, we were interested in investigating DD relationships, which represent an unprecedented opportunity to exploit mechanistic knowledge and repurpose treatments from one disease to the other. We thus proposed a computational framework for the application of MF to the study of DD relationships. Considering the inverse comorbidity between lung cancer (LC) and Alzheimer’s disease (AD) as a case study, different methodologies for the orientation of the metagenes were tested and the “disease-pointing” one, orientating metagenes based on the case vs. control behavior of metasamples, was shown to give better performance.

The framework proposed here and applied to the study of inverse comorbidity between LC and AD can be used in the future to investigate direct/inverse comorbidity relations among other combinations of diseases. More complex patterns of direct and/or inverse comorbidities, involving more than two diseases, could also be studied. Moreover, we chose sICA as our MF algorithm and we employed transcriptomic data. However, the framework proposed here can also be implemented with other MF approaches (e.g., NMF, PCA) or different omics data types (e.g., methylome, proteome). Moreover, given the growing availability of multi-omics data measured on the same set of patients, we envisage the possible use of our framework with multi-omics factors, obtained with approaches such as multi-omics factor analysis (MOFA) or tensorial ICA (tICA) [58,59].

Finally, we performed a functional analysis of the genes involved in the subnetwork containing negative links between AD and LC factors. Our results confirmed previously identified molecular mechanisms underlying this inverse comorbidity, such as the involvement of the immune system and mitochondrial processes, but also new candidate factors have been identified. Overall, our results suggest that the MF RBH-based extended approach can be of biological and medical relevance when investigating the molecular bases of DD relationships.

## 4. Materials and Methods

### 4.1. Data Collection

Three microarray datasets from NCBI Gene Expression Omnibus (GEO) (https://www.ncbi.nlm.nih.gov/geo/) were collected for Alzheimer’s disease: GSE4757 and GSE48350 obtained from four brain regions: the hippocampus, entorhinal cortex, superior frontal cortex, post-central gyrus and GSE5281 obtained from six brain regions: the entorhinal cortex, hippocampus, medial temporal gyrus, posterior cingulate, superior frontal gyrus and primary visual cortex. The last two datasets were split based on the region of the brain in which the samples were collected, obtaining a total of 11 AD datasets composed of both case and control samples. Concerning lung cancer, three microarray datasets from NCBI GEO were collected: GSE19188, GSE19804 and GSE33532. The last one, involving four biopsies from the same sample, was split into four datasets. We thus obtained a total of six LC datasets composed of case and control samples. All these microarray datasets were obtained with the same microarray platform, the Affymetrix HG U133Plus2, to reduce bias due to inter-platform differences. Moreover, each study was normalized separately using the frozen multiarray analysis (fRMA) [60]. Additionally, the RNA-seq lung dataset downloaded from The Cancer Genome Atlas (TCGA; https://tcga-data.nci.nih.gov/tcga/) was added to the analysis. In this case, RSEM normalized Level 3 data were downloaded from TCGA. Genes having zero values in more than 30% of samples were filtered out and the data were then log2 transformed.

### 4.2. Biological Characterization of the Communities

We characterized the communities obtained in the –RBH AD/LC subnetwork using the following annotations: MSigDB signatures [33], microenvironment cell populations counter signatures [34], predefined TCGA lung cancer subtypes [35] and the metagenes computed in [27], here referred to as CIT. Concerning subtypes association, we employed the metasamples obtained from the TCGA lung cancer data.

We tested the significance of the association with the predefined LC subtypes by performing a two-sided Wilcoxon test (cancer subtype vs. all other samples) and corrected for multiple testing using Bonferroni. For all other biological annotations involving genes, we employed the metagenes contained in each community. We associated to each community of the –RBH AD/LC subnetwork a “consensus metagene” corresponding to the average of all the metagenes contained in the community, paying attention to first concordantly orientate all the metagenes of the community based on the signs of their correlations (all the metagenes in the community were oriented based on the direction of LC). We then defined the top-contributing genes of a community as those genes having a weight in the consensus metagene higher than three standard deviations in absolute value. The top-contributing genes were then divided into “up” and “down” based on their sign in the consensus metagene and tested for their intersection with the various collections of signatures. For cell type specific signatures, we used a Fisher’s exact test with Bonferroni correction; for MsigDB, we employed its default enrichment test [33].

After testing the association of each community with all the considered annotations (MSigDB signatures, MCP counter cell type signatures, the lung cancer subtypes available for TCGA data and the CIT metagenes), we associated the annotation that is more consistently found across the different tests to each community. In Appendix A, the annotations associated to each community together with their associated *p*-values are more extensively described.

Finally, to test the reproducibility of the identified consensus metagenes in other cancers, we used the metagenes computed with sICA on TCGA transcriptomics data from 32 different cancer types [28]. Then, for each community in the –RBH AD/LC subnetwork, we computed the number of cancers having at least one correlated metagene. The resulting values are reported in Appendix A.

## Figures and Tables

**Figure 1 ijms-20-03114-f001:**
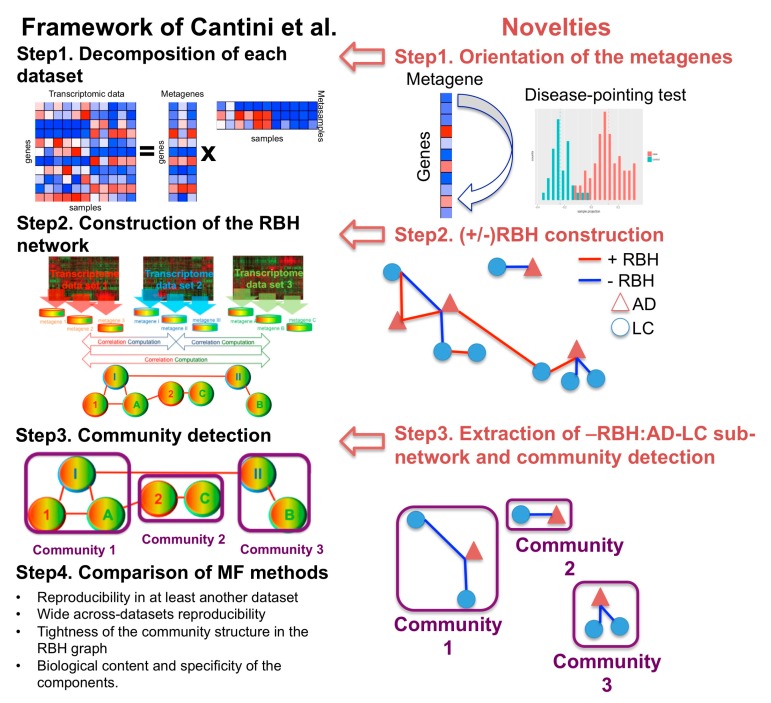
Schematic view of the framework and the novelties introduced with respect to [12]. RBH = reciprocal best hit. MF = matrix factorization. AD = Alzheimer’s disease. LC = lung cancer.

**Figure 2 ijms-20-03114-f002:**
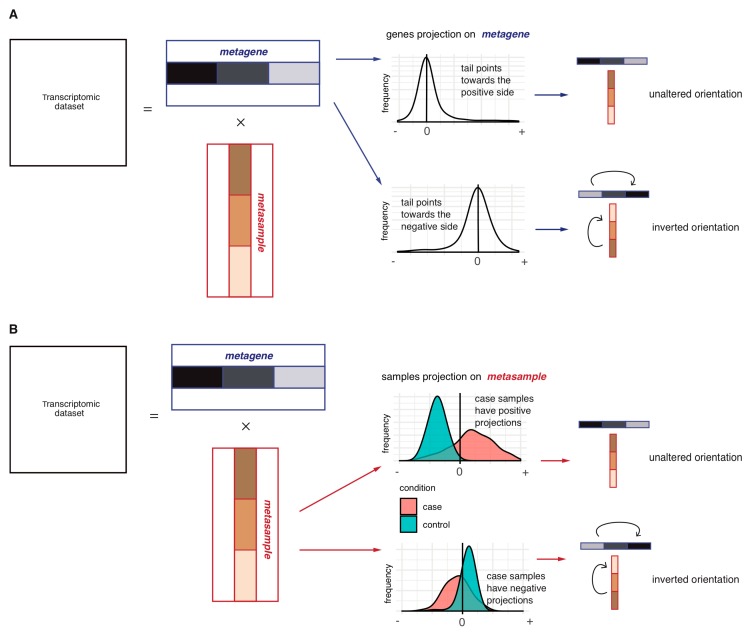
Schematic representation of the metagene orientation procedures: Long-tail-pointing (**A**) and disease-pointing (**B**). In (**A**), each couple (metagene, metasample) is oriented based on the distribution of the metagene weights. Two possible scenarios may thus be verified depending on the distribution of the weights constituting the metagene: if the long tail of the distribution is already in the positive side, no operation is performed; if instead the long tail is pointing in the negative direction, the orientation of the two vectors is inverted. In (**B**), each couple (metagene, metasample) is oriented based on the distribution of the metasample weights. Two possible scenarios may thus be verified depending on the case vs. control distribution of the weights constituting the metasample: if case samples have higher weights with respect to control samples, no operation is performed; in the opposite case, the two vectors are inverted.

**Figure 3 ijms-20-03114-f003:**
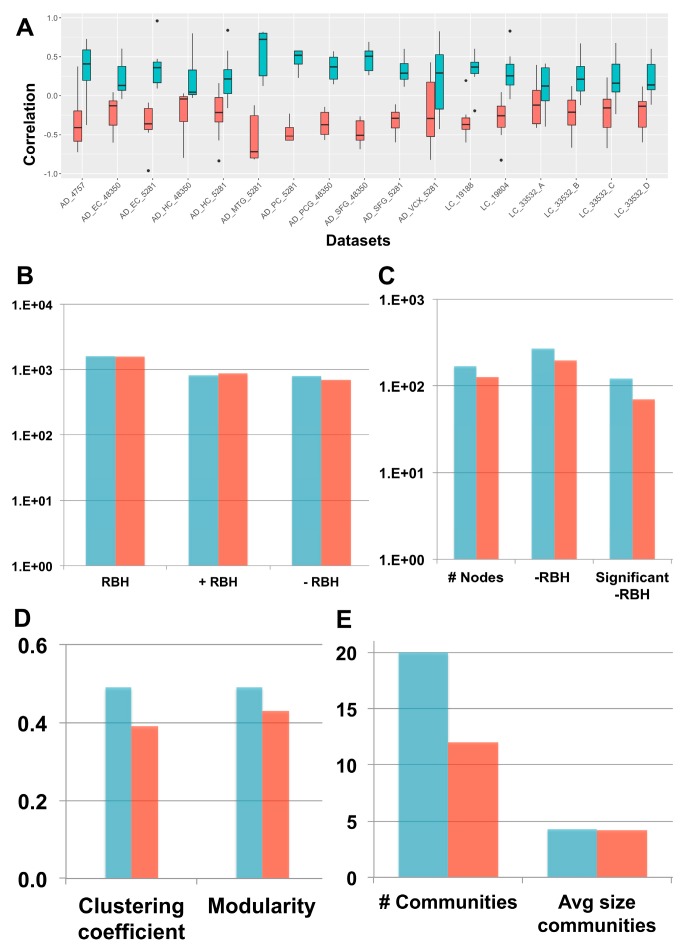
“Long-tail-pointing” (red) vs. “disease-pointing” (blue) orientation of the stabilized independent component analysis (sICA) factors. (**A**) The two methods of factor orientation are compared based on the correlation of the obtained metagenes with the case vs. control genes’ fold change. (**B**) The two methods are compared based on the number of links present in their RBH network. Total RBHs (RBH), positive RBHs (+RBH), negative RBHs (–RBH). (**C**–**E**) The two methods are compared based on the structure of their –RBH AD/LC subnetwork, relevant for the study of inverse comorbidity. In (**C**), the number of nodes and links in the subnetwork are compared. Number of nodes (# Nodes), negative RBHs connecting an AD component with a LC component (–RBH) and negative RBHs connecting an AD component with a LC component that are associated to nodes with significant differential behaviour (Wilcoxon *p*-value < 0.05) between case and control (significant –RBH). In (**D**), the clustering coefficient and modularity of the subnetwork are considered. In (**E**), the number of communities and their average size in the subnetwork is taken into account.

**Figure 4 ijms-20-03114-f004:**
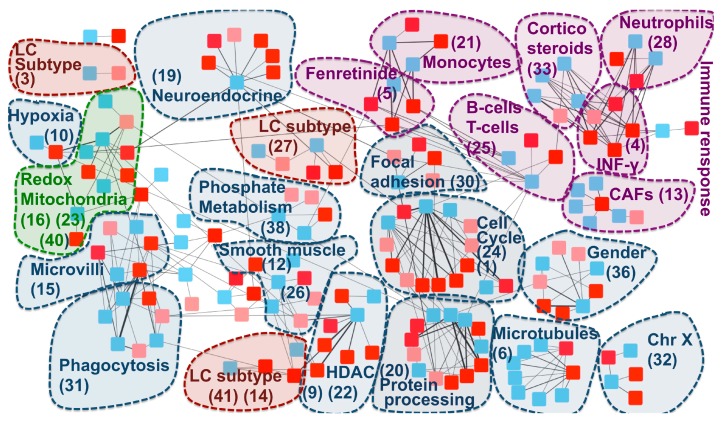
–RBH AD/LC subnetwork with biological annotations. Each node in the network corresponds to a metagene; the list of metagenes associated to each community ID is reported in Appendix A. Colours are linked to the diseases: red for AD and blue for LC. In AD, datasets obtained from the same region of the brain are denoted with different shades of red (normal and light red). The nodes are organized into communities. Each community is denoted with a number corresponding to its ID and the main biological annotation associated to them (see Appendix A for an extensive report). HDAC = histone deacetylase.

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
