# Peer review of "Molecular Inverse Comorbidity between Alzheimer’s Disease and Lung Cancer: New Insights from Matrix Factorization"

_ijms, 2019, doi:10.3390/ijms20133114_

Reviewer 1 Report

This manuscript provides another supporting evidence of the known inverse comorbidity between Alzheimer’s disease and lung cancer, using an improved method.

The results are shown with adequate depth of Discussion. However, this manuscript may be missing (or may have dropped) some details. Elaborating on the details of the methodology would help making this manuscript as a model framework for other similar analyses.

Below are my comments.

The authors have claimed the approach of the ‘disease-pointing’ orientation of metagenes as a novelty in this manuscript. So, it would be better if the description of ‘disease-pointing’ orientation could be elaborated. Some illustrations would help.

Regarding the calculation of RBH, I wonder if it can incorporate a cut-off value. Correlation coefficient that is near to 0, regardless of the sign, may represent no to a very little correlation. Also, it is not clear whether the level of correlation (i.e., strong or weak) was taken into account for the community identification. If so, please elaborate the related parts in Methods and/or Results section.

Regarding the metagenes referred to as ‘CIT’, my understanding is that CIT is a series of dataset about bladder cancer, from which the components (or metagenes) CIT-# were identified in the cited paper. I wonder if the authors meant to refer to the metagenes on lung cancer that was also identified in the same paper. If not, what is the rationale behind associating lung cancer metagenes with bladder cancer metagenes? Please clarify.

In section 2.2, it would be useful to provide a ballpark summary on the identified communities and how the community IDs were assigned, before starting to use community IDs in the following sections.

Regarding the data collection, although the datasets were collected from GEO and TCGA, there is no mention of how they were processed, including normalization and correcting for the batch effects, which would be non-trivial given that each dataset is from separate studies, especially between microarray-based expression datasets and RNA-seq based expression datasets. Maybe, normalization is not critical for this study, but it is worth mentioning.

Also, please clarify on the method for gene expression fold-change calculation.

Author Response

Reviewer 1:

This manuscript provides another supporting evidence of the known inverse comorbidity between Alzheimer’s disease and lung cancer, using an improved method.

The results are shown with adequate depth of Discussion. However, this manuscript may be missing (or may have dropped) some details. Elaborating on the details of the methodology would help making this manuscript as a model framework for other similar analyses.

We thank the Reviewer for his/her careful reading of the manuscript. Indeed, as noted by the Reviewer, this work is built from previous analyses on molecular inverse comorbidities. We might have used these previous analyses too much as references and failed to clearly the methodology behind the work presented in the current manuscript. In order to correct this point, we now added more details in the Results section 2.1 describing the framework of our analysis, and we added a new figure (current Figure 2) to better clarify the disease-pointing and long-tail pointing orientation.

 In addition to these enhanced descriptions of our protocol, we now also provide the code to run our framework on other datasets. This code is available on github at https://github.com/agreco92/inverse_comorbidity_MF

 Below are my comments.

1. The authors have claimed the approach of the ‘disease-pointing’ orientation of metagenes as a novelty in this manuscript. So, it would be better if the description of ‘disease-pointing’ orientation could be elaborated. Some illustrations would help.

We thank the Reviewer for this comment, as indeed, in  the first version of the manuscript both long-tail-pointing and disease-pointing orientations were just briefly explained. We have now enriched these explanations with further details (see results section 2.1.1), and added a new figure,  Figure 2 to graphically clarify how we perform the orientation of the metagenes proposed in this work.

2. Regarding the calculation of RBH, I wonder if it can incorporate a cut-off value. Correlation coefficient that is near to 0, regardless of the sign, may represent no to a very little correlation. Also, it is not clear whether the level of correlation (i.e., strong or weak) was taken into account for the community identification. If so, please elaborate the related parts in Methods and/or Results section.

We agree with the Reviewer that correlation values near to zero should not be taken into account, and in this regard, the RBH calculation can incorporate a cutoff value.

At the same time, it should be noted that the RBH approach does not correspond to a simple correlation analysis. As described by equations (1) and (2) in the results section 2.1.2, two metagenes Mi and Nj are linked by an RBH if and only if their correlation is maximal in respect to both the correlations of  Mi with all other metagenes obtained from dataset TN and the correlations of Nj with all other metagenes obtained from dataset TM. For defining a +RBH (equation (1)) positive correlations are considered, while for –RBH (equation (2)) negative correlations are taken into account. Such a procedure intrinsically selects high correlations and originates a sparse graph thus not requiring further filtering of the identified links. A comparison of a correlation-based network vs. RBH network can be found in the Supplementary Figure S1 in our previous publication (Cantini, L., Kairov, U., De Reyniès, A., Barillot, E., Radvanyi, F., & Zinovyev, A. (2019). Assessing reproducibility of matrix factorization methods in independent transcriptomes). In order to clarify this point, we have now improved the description of the RBH procedure in the Results section 2.1.2.

Finally, the correlation value associated to the RBH links is taken into account in the community detection, as a standard procedure of the MCL algorithm. We now specify this point in the Results section 2.1.3.

3. Regarding the metagenes referred to as ‘CIT’, my understanding is that CIT is a series of dataset about bladder cancer, from which the components (or metagenes) CIT-# were identified in the cited paper. I wonder if the authors meant to refer to the metagenes on lung cancer that was also identified in the same paper. If not, what is the rationale behind associating lung cancer metagenes with bladder cancer metagenes? Please clarify.

The Reviewer is correct: the CIT metagenes have been obtained from bladder cancer data and the main focus of the cited paper {Biton, A., Bernard-Pierrot, I., Lou, Y., Krucker, C., Chapeaublanc, E., Rubio-Pérez, C., ... & Grieco, L. 2014. Independent component analysis uncovers the landscape of the bladder tumor transcriptome and reveals insights into luminal and basal subtypes. Cell reports, 9(4), 1235-1245} is indeed bladder cancer. However, in the same paper, the correlation of the CIT metagenes with metagenes obtained from other cancer types is also studied. As shown in Figure 2 of that manuscript, the CIT metagenes are reproduced in other cancers. The most evident case of reproducibility of CIT metagenes in other cancers is represented by those metagenes reflecting basic cellular functions, such as proliferation, that are found in the majority of the cancers considered in the study. In following works such as {Kairov, U., Cantini, L., Greco, A., Molkenov, A., Czerwinska, U., Barillot, E., & Zinovyev, A. (2017). Determining the optimal number of independent components for reproducible transcriptomic data analysis. BMC genomics, 18(1), 712} and {Cantini, L., Kairov, U., De Reyniès, A., Barillot, E., Radvanyi, F., & Zinovyev, A. 2019. Assessing reproducibility of matrix factorization methods in independent transcriptomes}, we already employed CIT metagenes for interpretation of the metagenes present in 32 cancer types.

As shown in Supp Table 2 , the results obtained through correlation with CIT metagenes  are confirmed by MsigDB or MCP counter signatures enrichment. Moreover, the CIT metagenes that we find correlated with our communities are CIT3, CIT4, CIT7, CIT8 and CIT18 corresponding to smooth muscle, mitochondria, cell cycle, Immune and neuroendocrine tumors all signals that make sense in the context of LC-AD inverse comorbidity.

4. In section 2.2, it would be useful to provide a ballpark summary on the identified communities and how the community IDs were assigned, before starting to use community IDs in the following sections.

Community IDs have been assigned as in the output generated by MCL and they are only a way to univocally flag the communities present in the RBH. A summary of the communities present in the RBH network is provided in Supp Table 2. In the previous version of the paper we referred to this table only in section 2.3, we now refer to it already in section 2.2.

5. Regarding the data collection, although the datasets were collected from GEO and TCGA, there is no mention of how they were processed, including normalization and correcting for the batch effects, which would be non-trivial given that each dataset is from separate studies, especially between microarray-based expression datasets and RNA-seq based expression datasets. Maybe, normalization is not critical for this study, but it is worth mentioning.

Concerning the data of lung cancer and Alzheimer’s disease obtained with microarrays, all of them have been selected from the same microarray platform, the Affymetrix HG U133Plus2, to reduce biases due to inter-platform differences. Moreover, each study was normalized separately using the frozen Robust Multiarray Analysis (fRMA) {McCall, M. N., Jaffee, H. A., & Irizarry, R. A. (2012). fRMA ST: frozen robust multiarray analysis for Affymetrix Exon and Gene ST arrays. Bioinformatics, 28(23), 3153-3154.}. The RNA-seq Lung dataset downloaded from the Cancer Genome Atlas (TCGA; https://tcga-data.nci.nih.gov/tcga/) was also added to the analysis. In this case RSEM normalized Level 3 data were downloaded from TCGA. Genes having zero values in more than 30% of samples were filtered out and the data were then log2 transformed.

Concerning the correction for batch effects, no correction was performed given that each dataset was separately decomposed with MF. Moreover,  the advantage of using sICA is also in its capability to separate effects such as experimental noise or batch effects from the biological signals. So if batches are present in the data the sICA output will contain a component associated to batches. In order to clarify our manuscript, we now have extended the method descriptions in section 4.1.

6. Also, please clarify on the method for gene expression fold-change calculation.

We now detail the fold-change computation at line 238: “fold change measures expression changes between two states (case vs control in our case) and for each gene in the log2-transformed expression matrix it is computed as avg(case samples)-avg(control samples).

Reviewer 2 Report

This paper developed a new implementation of MF. This method was applied for inversing comorbidity association between Alzheimer’s disease (AD) and lung cancer (LC). I have the following concerns:

· The paper does not explain clearly its advantages with respect to the literature: it is not clear what is the novelty and contributions of the proposed work: does it propose a new method? Or does the novelty only consists in the application? therefore these paper should be discussed

Molecular evidence for the inverse comorbidity between central nervous system disorders and cancers detected by transcriptomic meta-analyses

Molecular Evidence for the Inverse Comorbidity between Central Nervous

Hybrid resampling and multi-feature fusion for automatic recognition of cavity imaging sign in lung CT

Glioma Segmentation Using a Unified Algorithm in Multimodal MRI Images

·  The advantage of the proposed method with respect to other methods in the literature should be clarified.

· Quality of figures is so important too. Please provide some high-resolution figures. Some figures have a poor resolution

· Conclusion should state scope for future work.

· Results need more explanations. Additional analysis is required at each experiment to show the its main purpose.

· Need detailed explanation of the preprocessing steps.

Author Response

Reviewer 2:

This paper developed a new implementation of MF. This method was applied for inversing comorbidity association between Alzheimer’s disease (AD) and lung cancer (LC). I have the following concerns:

We thank the Reviewer for his comments. Below we report the point-by-point answer to his/her questions. Additionally, at https://github.com/agreco92/inverse_comorbidity_MF we now provide the code to run our framework on other datasets.

1. The paper does not explain clearly its advantages with respect to the literature: it is not clear what is the novelty and contributions of the proposed work: does it propose a new method? Or does the novelty only consists in the application? therefore these paper should be discussed

             Molecular evidence for the inverse comorbidity between central nervous system disorders and cancers detected by transcriptomic meta-analyses

            Molecular Evidence for the Inverse Comorbidity between Central Nervous

            Hybrid resampling and multi-feature fusion for automatic recognition of cavity imaging sign in lung CT

            Glioma Segmentation Using a Unified Algorithm in Multimodal MRI Images

The work proposes an adaptation of a pre-existing method to the study of a new biological problem, that is the study of  Disease-Disease (DD) relationships. The work thus contains novelties both on the methodological and biological side.

Concerning the methodology, in a previous work (Cantini, Laura, et al. "Assessing reproducibility of matrix factorization methods in independent transcriptomes." (2019).) we designed a metric to infer univocal correspondences between the metagenes obtained by an MF algorithm on multiple independent datasets profiled from the same biological condition (e.g. same cancer tissue), and used this metric to design a methodological framework that revealed relevant pathways characteristic of colorectal cancer. In the current work, we extend this previously defined MF framework for the study of  Disease-Disease (DD) relationships. For example, given the existence of positive and negative DD relationships, we adapted the framework to distinguish molecular relationships concordantly and discordantly altered in datasets coming from different diseases. All the methodological novelties introduced to the previous framework are summarized in Figure 1.

On the biological side, we instead summarize all the new information we obtained into the Results section 2.3.

Concerning then the advantages of our method in respect to the existing literature, we summarized them in the beginning of the fourth paragraph in the introduction. We have now modified the text to make this point more clear (see line 84):“ Recently, transcriptomic meta-analyses revealed sets of significantly up and down regulated genes that are shared across diseases displaying different patterns of direct and inverse comorbidities [25,26]. We hereby propose to use an MF approach to study the molecular bases of DD relationships. In fact previous methods based on differential expression analysis only focuses on the predominant signals present in the data, failing to capture alternative signals and local behaviors [3]. These limitations are overcome by MF that learns metagenes, i.e. ranking of genes, without focusing on single sets of predominant genes. Moreover, contrarily to differential expression analysis, MF jointly provides metagenes and metasamples, i.e. also grouping samples together with their biological characterization. However, applying MF to the study of the molecular bases of DD relationships requires innovative adaptations. We thus propose to methodologically extend our previously defined MF framework for the particular study of DD relationships [12]. Moreover, given the existence of positive and negative DD connections, we also adapt the framework to distinguish molecular relationships concordantly and discordantly altered in datasets coming from different diseases. ”

Concerning the citations that the Reviewer proposed to add, the first one corresponds to a paper that was already cited in our manuscript (see citation 25 ), and we were not able to identify the second one. We carefully checked the two other papers proposed by the Reviewer (Hybrid resampling and multi-feature fusion for automatic recognition of cavity imaging sign in lung CT; Glioma Segmentation Using a Unified Algorithm in Multimodal MRI Images), but did not identify the relationship with our  work, neither from a biological nor methodological point of view. Indeed, they do not make use of Matrix Factorization and they do not concern AD-LC inverse comorbidity or Disease-Disease relations more in general.

2.  The advantage of the proposed method with respect to other methods in the literature should be clarified.

As mentioned above, such advantages are already mentioned in the fourth paragraph of the introduction: “ Recently, transcriptomic meta-analyses revealed sets of significantly up and down regulated genes that are shared across diseases displaying different patterns of direct and inverse comorbidities [25,26]. However, differential expression analysis only focuses on the predominant signals present in the data, failing to capture alternative signals and local behaviors [3]. These limitations are overcome by MF that learns metagenes, i.e. ranking of genes, without focusing on single sets of predominant genes. Moreover, contrarily to differential expression analysis, MF jointly provides metagenes and metasamples, i.e. also grouping samples together with their biological characterization. We hereby propose to use an MF approach to study the molecular bases of DD relationships. This, however, requires innovative adaptations. ”

The cited methods, represent to our knowledge the only computational works performed to investigate the molecular bases of direct or inverse comorbidities.

3. Quality of figures is so important too. Please provide some high-resolution figures. Some figures have a poor resolution

Thanks for pointing that out, we now upload the Figures in higher quality.

4.  Conclusion should state scope for future work.

We clarified the future perspectives for the use of our framework in the study of other disease-disease relationships and/or in its combined use with multi-omics analysis tools.

5. Results need more explanations. Additional analysis is required at each experiment to show the its main purpose.

We now better explained the steps of our framework and the methodological novelties that we here introduced highlighting their main purpose. A new Figure 2 has now also been provided to better explain the new procedure for the orientation of the metagenes proposed in this work. Additionally we now provide the code to apply the framework to new data.

6. Need detailed explanation of the preprocessing steps.

Concerning the data of lung cancer and Alzheimer’s disease obtained with microarrays, all of them have been obtained with the same microarray platform, the Affymetrix HG U133Plus2, to reduce bias due to inter-platform differences. Moreover, each study was normalized separately using the frozen Robust Multiarray Analysis (fRMA) (McCall, M. N., Jaffee, H. A., & Irizarry, R. A. (2012). fRMA ST: frozen robust multiarray analysis for Affymetrix Exon and Gene ST arrays. Bioinformatics, 28(23), 3153-3154.). Additionally, the RNA-seq Lung dataset downloaded from the Cancer Genome Atlas (TCGA; https://tcga-data.nci.nih.gov/tcga/) was added to the analysis. In this case RSEM normalized Level 3 data were downloaded from TCGA. Genes having zero values in more than 30% of samples were filtered out and the data were then log2-transformed. We now added these details in methods, section 4.1.

Round  2

Reviewer 1 Report

Regarding the fold-changes in gene expression, the fold-change estimation was done in a quite simple way, not by a proper package, such as EB-seq if RSEM was used for quantification, which is still relatively simple to do. Also, the range of fold-change that can be detected from an microarray-based transcriptome data and from an RNA-seq transcriptome data is quite different. RNA-seq data is a lot more sensitive than microarray data in that regard, which could cause some inconsistency between different datasets of the same tissue/disease for the same gene.

Having said that, as fold-change is only supplementary for inferring DD relationship, those concerns may not be so critical. However, this would be something to consider for further improvement or for related analysis in the future.

Other than this, I have a few more minor things.

In line 171, “and a RBH network, called in here and in the following RBH network,”, is anything missing in this part of the sentence?

In lines 177-179, the sentence (“The improved RBH-based framework ... described above.”) is unclear. Maybe, you could paraphrase it?

In section 2.1.1 and Figure 2, it appears that the strength of correlation was used as ‘weight’, but it was not clearly defined.

There are other sentences that do not read well or unclear. Please give it a careful proofreading.

Author Response

We thank the Reviewer for his/her careful reading of the manuscript. Below the point-by-point answer to his/her comments.

Regarding the fold-changes in gene expression, the fold-change estimation was done in a quite simple way, not by a proper package, such as EB-seq if RSEM was used for quantification, which is still relatively simple to do. Also, the range of fold-change that can be detected from an microarray-based transcriptome data and from an RNA-seq transcriptome data is quite different. RNA-seq data is a lot more sensitive than microarray data in that regard, which could cause some inconsistency between different datasets of the same tissue/disease for the same gene.

Having said that, as fold-change is only supplementary for inferring DD relationship, those concerns may not be so critical. However, this would be something to consider for further improvement or for related analysis in the future.

We want first to stress that fold-change is not part of our RBH framework. We only use it in this work with the aim of comparing the performances of the two procedures for the orientation of metagenes: “disease-pointing” and “long-tail pointing”. 

Additionally, the AD and LC datasets to which we applied the fold-change for the “disease-pointing” vs. “long-tail pointing” comparison are all datasets obtained with the microarray technology. The only RNA-seq dataset employed in the analysis is the TCGA LUAD dataset, that is constituted by only cancer samples. We thus did not apply our orientation procedure to this dataset and we only employed it to test associations between the communities identified in the AD/LC –RBH subnetwork and predefined Lung cancer subtypes. We thus not have problems connected to instabilities of the fold-change in RNA-seq data and we do not need a formulation of the fold-change that is appropriate for RSEM RNA-seq data.

Moreover, the choice to consider the gene expression case vs. control fold-change for the comparison of “disease-pointing” vs. “long-tail pointing” is justified by the fact that to associate a metagene to a specific biological function or pathway, we need to perform enrichment tests using databases of functional annotations (e.g. Reactome, GO). When dealing with comorbidities it is important to not only identify the pathway/function associated to each metagene, but also to infer the sign of activity of such pathways/functions (e.g. up-regulated  in LC and down-regulated in AD). This task can be easily achieved once the metagenes are positively correlated with the gene fold-change.

We are thus interested in the sign of the fold-change and we want in particular to have metagenes whose sign is concordant with the sign of the gene expression fold-change. Having a different formulation of the fold-change, that should not alter its sign, would thus not affect the results presented in the paper. Indeed, when comparing “disease-pointing” vs. “long-tail pointing” orientation, we are dealing with exactly the same metagenes and their only difference is in their orientation/sign. As a result, a different fold-change computation would equally affect the correlation between the metagenes of “disease-pointing” and “long-tail pointing” and the only thing that matters for our results reported in Figure 3A is the concordance between the sign of the metagene and the sign of the fold-change. 

Other than this, I have a few more minor things.

In line 171, “and a RBH network, called in here and in the following RBH network,”, is anything missing in this part of the sentence?

If the reviewer is referring to the current line 121, there was maybe some problem with the processing of the document with track changes, the text is indeed:

“using the RBH metric, links between metagenes are inferred and a network, called here and in the following RBH network, is constructed”

which refers to the fact that using the RBH metric we built a network that we call RBH network.

In lines 177-179, the sentence (“The improved RBH-based framework ... described above.”) is unclear. Maybe, you could paraphrase it?

We modified the sentence corresponding to current lines 128-129. The current version is:

“The improved RBH-based framework here proposed maintains the three sequential steps described above.”

In section 2.1.1 and Figure 2, it appears that the strength of correlation was used as ‘weight’, but it was not clearly defined."

According to the definition of +RBH and -RBH (see formula 1 and 2 at lines 183-185) two metagenes are linked if their correlation is maximal. This correlation value is used as weight for the links of the RBH network. We better clarify this point at line 197. Additionally, we now specify at line 186 that the correlation used in the paper is Spearman correlation.

Reviewer 2 Report

no more comments

Author Response

We thank the Reviewer for his/her careful reading of the manuscript and his/her comments.